# DNN Quantization with Attention

## Abstract

Low-bit quantization of network weights and activations can drastically reduce the memory footprint, complexity, energy consumption and latency of Deep Neural Networks (DNNs). Many different quantization methods like min-max quantization, Statistics-Aware Weight Binning (SAWB) or Binary Weight Network (BWN) have been proposed in the past. However, they still cause a considerable accuracy drop, in particular when applied to complex learning tasks or lightweight DNN architectures. In this paper, we propose a novel training procedure that can be used to improve the performance of existing quantization methods. We call this procedure DNN Quantization with Attention (DQA). It relaxes the training problem, using a learnable linear combination of high, medium and low-bit quantization at the beginning, while converging to a single low-bit quantization at the end of the training. We show empirically that this relaxation effectively smooths the loss function and therefore helps convergence. Moreover, we conduct experiments and show that our procedure improves the performance of many state-of-the-art quantization methods on various object recognition tasks. In particular, we apply DQA with min-max, SAWB and BWN to train 2bit quantized DNNs on the CIFAR10, CIFAR100 and ImageNet ILSVRC 2012 datasets, achieving a very good accuracy comparing to other conterparts.

## 1   Introduction and Related Work

In the last decade, Deep Neural Networks (DNNs) in general and Convolutional Neural Networks (CNNs) in particular became state-of-the-art in many computer vision tasks, such as image classification or segmentation, object detection and face recognition (LeCun et al. (1998); Iandola et al. (2016); Simonyan & Zisserman (2014); Graham (2014); Szegedy et al. (2015)). However, to be the state-of-the-art, DNNs often contain a large number of trainable parameters and require considerable computational power. Therefore, due to their large power and memory consumption, implementing DNNs on embedded systems with limited resources can be a real challenge. To alleviate this problem, a large number of different network compression methods that reduce the resource requirements of DNNs has been proposed in the past. Among them are for example pruning, distillation or quantization methods.

Pruning methods have been first introduced by LeCun et al. (1990). They identify and remove the most insignificant DNN parameters and yield networks with a reduced memory footprint and a smaller computational complexity. As reported by Yamamoto & Maeno (2018); Ramakrishnan et al. (2020); He et al. (2020), pruning methods can either remove single DNN parameters, intermediate inputs or even whole network layers that are, according to a specific criteria, irrelevant for a good network performance.

Another line of work is distillation. As introduced by Hinton et al. (2015), it aims at training a small student DNN to reproduce the output of a bigger teacher network. While distillation methods initially only matched the final outputs of the teacher and student networks, methods evolved to take into account intermediate representations (Romero et al. (2014); Koratana et al. (2018)).

Our work is about DNN quantization, where weights and activations are represented with a smaller number of bits $n << 32$. Quantization reduces the memory footprint of DNNs

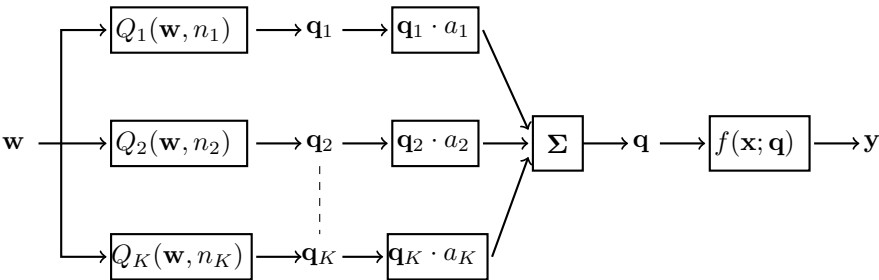

Figure 1: Quantizing the weights of a single network layer, using a linear combination of $K$ different quantizers $\mathbf{Q} = \{Q_1, Q_2, \cdots, Q_K\}$. Note, that each quantizer uses a different bitwidth, i.e., $n = \{n_1, n_2, \cdots, n_K\}$. The resulting quantized weights $\mathbf{q}_k$ are multiplied with attention values $a_k \in [0, 1]$ that reflect the importance of the corresponding quantization function $Q_k(\cdot)$. The attention values are optimized during training according to algorithm 1).

because the number of bits that are required to store their parameters is reduced. However, it also reduces their computational complexity, because low bit operations can be implemented more efficiently on dedicated low precision hardware (Merolla et al. (2014); Farabet et al. (2011); Cowan et al. (2020); Han et al. (2020); Hacene et al. (2018)).

Following Gupta et al. (2015), many works have experimentally demonstrated that neural networks do not lose a lot of performance when their parameters are restricted to a small set of possible values. For instance, Choi et al. (2018) introduced PArameterized Clipping ac-Tivation (PACT) in combination with Statistics-Aware Weight Binning (SAWB) –a method that aims at uniformly quantizing both weights and activations to $n$ bit. Learned Step Size Quantization (LSQ) presented by Esser et al. (2019) is a quantization method that learns the quantization steps during training. Unlike other methods, it scales the gradient during backpropagation to speed up the training. Gradient scaling is important especially at transition points.

Nikolić et al. (2020) proposed Bit-Pruning, a quantization method to learn the number of bits that each layer requires to represent its parameters and activations. In the same vein, Differentiable Quantization of Deep Neural Networks proposed by Uhlich et al. (2019) (DQDNN) tries to combine the features of both LSQ and Bit-Pruning. They propose a quantization technique where both the number of bits and the quantization steps are learned simultaneous. Other more aggressive quantization methods proposed to use low-bit precision down to binarization (resp. ternarization) with only two (resp. three) possible values and one (resp. two) bit storage for each parameter and/or activation (Hubara et al. (2016); Courbariaux et al. (2015); Li et al. (2016b); Zhu et al. (2016); Li et al. (2016a)).

Zhou et al. (2017) observed that training quantized networks to low precision benefits from incremental training. Rather than quantizing all the weights at once, they are quantizing them incrementally in groups, with some training iterations between each quantization step. In practice, 50% of the weights are quantized in the first step, then 75%, 87.5% and finally 100%. Another method for incremental quantization is Binary-Relax (BR) (Yin et al. (2018)). Rather than splitting the parameters into groups, it uses a weighted linear combination of quantized and full-precision parameters and adapts a strategy to push the weights towards the quantized state, by gradually increasing the scaling factor corresponding to the quantized parameters. However, their strategy how to move from full-precision to quantized parameters is handcrafted and may not be optimal.

In comparison to most of the previously mentioned works, we do not propose an improved quantization method, but a way how to train a quantized DNN with any existing quantizer. We rely on the fact that the DNN performance can be increased if the network has the ability to learn other features in addition to its own parameters. In the context of DNN quantization, this has already been discussed by Elsken et al. (2019); Ramakrishnan et al. (2020); Hacene et al. (2019); Uhlich et al. (2019); Nikolić et al. (2020). In this contribution, we introduce DNN Quantization with Attention (DQA), an attention mechanism-based learnable approach for DNN quantization (Vaswani et al. (2017)). As shown in Figure 1,

rather than quantizing with just a single bitwidth DQA performs quantization with multiple different bitwidths in parallel. At each stage of the training, DQA can select the quantizer with the optimal bitwidth by putting more attention weight on it. In particular, by starting with a uniform initialization of the attention weights at the beginning of the training, DQA can move smoothly from high precision to low precision quantization. We demonstrate in this paper that this learnable approach results in better DNN accuracy for the exact same complexity and number of bits.

The outline of the paper is as follows. In Section 3 we introduce the proposed method. Section 4 presents experiments results and compares our method with other state-of-the-art approaches on challenging computer vision datasets. Finally, we conclude in Section 5.

## 2    Background

In the following, we explain how differentiable quantization with attention (DQA) can be used to train quantized DNNs. For simplicity reasons, we only discuss how to train DNNs with quantized weights. However, DQA is general and can also be used to train DNNs with quantized weights and activations. In particular, we will also provide experimental results for this case in Section 4.

Training DNNs with quantized weights is challenging. Especially when considering low bitwidths $n << 32$, quantization can cause a severe accuracy degradation if compared to the full precision networks. This is mainly caused by the capacity reduction and by additional optimization issues that go hand in hand with the quantization. In particular, quantization yields non-smooth loss functions with gradients that are zero almost everywhere. As discussed by Uhlich et al. (2019), that effectively stops gradient backpropagation and therefore harms training.

For these reasons, quantized DNNs are usually not trained with standard gradient based training procedures, but require some tricks that allow for gradient backpropagation and that stabilize the training. The most commonly used trick is to apply straight through gradient estimates (STE) and to ignore the quantization during backpropagation. STE yield non-zero gradients that are suited for DNN training. However, at the same time they introduce a mismatch between the forward (FW) and backward (BW) pass, what often causes training instabilities and oscillations. More specifically, mismatch means that the gradients are calculated at the position of the quantized parameters, assuming the original float32 loss surface.

Note, that because of this FW/BW mismatch, training with STE does not necessarily converge to the optimum. This mismatch is for example problematic, if the gradient of the cost function changes signs within one quantization step. For this case, gradient descent with STE would start to oscillate near the quantization thresholds and would not converge to the optimum, even if the cost function is convex. Of course, this problem is most pronounced for low bitwidhts.

A training procedure that can alleviate this problem to some degree is Binary-Relax (BR). As proposed by Yin et al. (2018), BR does not only apply STE, but also uses a linear combination of the quantized and the full-precision parameters. This effectively reduces the FW/BW mismatch at the beginning, while still enabling the DNN to use the exact low-bit quantized loss at the end of the training.

## 3    Methodology

DQA builds on a similar approach. As shown in figure 1, let $f(\mathbf{x}; Q(\mathbf{w}; n))$ be the transfer function of a quantized DNN layer, where $\mathbf{x} \in \mathbb{R}^D$, $\mathbf{w} \in \mathbb{R}^M$ and $Q(\mathbf{w}; n)$ are the layer input, the full-precision weights and the quantized weights, respectively. Similar to the idea of Binary-Relax (BR), DQA relaxes the quantization problem and combines different quantization schemes during training. More specifically, instead of using just one single $Q(\mathbf{w}; n)$, we propose to train a quantized DNN with a set of $K$ different quantization functions that

are combined linearly during training as follows:

$$\mathbf{y} = f(\mathbf{x}; \mathbf{q}) \tag{1}$$

$$\mathbf{q} = \mathbf{Q}^T \mathbf{a} \tag{2}$$

$$\mathbf{Q} = \begin{bmatrix} Q_1(\mathbf{w}; n_1)^T \\ Q_2(\mathbf{w}; n_2)^T \\ \vdots \\ Q_K(\mathbf{w}; n_K)^T \end{bmatrix}, \tag{3}$$

where $\mathbf{q}$ is the weighted sum of $K$ quantized weight vectors, $\mathbf{Q} \in \mathbb{R}^{K \times M}$ is a matrix whose row vectors are the quantized weight vectors and $\mathbf{a} \in [0,1]^K$ is the attention vector on the quantization functions. Note, that each row of $\mathbf{Q}$ is calculated, using a different quantization function $Q_k(\mathbf{w}; n_k)$ and bitwidth $n_k \in \mathbb{N}$. In particular, we assume that the quantization functions in $\mathbf{Q}$ are sorted by the bitwidth, i.e., $n_1 < n_2 < ... < n_K$. In general, DQA is agnostic to the choice of the actual quantization method and can be used with any existing method like min-max, SAWB, binary or ternary quantization. In the following section, we review and define popular quantization methods that we used in our experiments.

The attention $\mathbf{a}$ is computed from a soft attention vector $\boldsymbol{\alpha} \in \mathbb{R}^K$, using a softmax function with temperature $T \in \mathbb{R}^+$, i.e.,

$$\mathbf{a} = \frac{e^{\frac{\boldsymbol{\alpha}}{T}}}{\sum_{k=1}^{K} e^{\frac{\alpha_k}{T}}}, \; \in \mathbb{R}^K. \tag{4}$$

In particular, $\mathbf{a}$ reflects the importance of the $K$ quantization methods $Q_k$. During training, the soft attention $\boldsymbol{\alpha}$ is treated as a trainable parameter that is optimized in parallel to the weights $\mathbf{w}$. Note that, increasing $\alpha_k$ will also increase the corresponding attention weight $a_k$ and therefore the importance of $Q_k(\mathbf{w}; n_k)$. In this manner, the quantized DNN can learn which bitwidth should be used at which stage, during the training.

DQA exponentially cools down the temperature $T$

$$T(b) = T(0)\Psi^b. \tag{5}$$

Here, $b = 1, 2, ..., B$ is the batch index for batch-wise training, $T(0) \in \mathbb{R}^+$ is the initial temperature and $\Psi \in [0, 1[$ is the decay rate. Because of that schedule, DQA progressively moves from the full mixture of quantization functions at the beginning of the training to just one single quantization function at the end of training.

Note, that BR can be seen as a special case of DQA, where $\mathbf{q} = [Q(\mathbf{w}; 2), Q(\mathbf{w}; 32)]^T$, i.e. for the case that we only use two quantizers with $n = 2$bit and $n = 32$bit, and for the case that we use a fixed schedule to change the attention vector $\boldsymbol{a}$. However, DQA has two advantages: 1) The way how we change $\boldsymbol{a}$ and move from high to low precision quantization is learned and data dependent. Hence, DQA can choose the optimal mixture of the quantizers at each training iteration. 2) As shown in Figure 2, DQA gives a smoother transition from high to low-precision parameters. Here, we plot the absolute quantization error for a fixed temperature based schedule for $\boldsymbol{a}$. In particular, we choose $\boldsymbol{\alpha} = [3/4, 1/4]^T$ and $\boldsymbol{\alpha} = [4/7, 2/4, 1/7]^T$ for BR and DQA with $2, 4, 8$bit quantization, respectively. Then, we start with a large temperature $T \to \infty$, for which we effectively take the average $\frac{1}{K} \sum_{k=1}^{K} Q_k(\mathbf{w}; n_k)$, and move towards 2bit quantization for $T \to 0$. Note, that for the whole interval that we consider for $T$, DQA results in a lower quantization error, meaning that it also yields a smaller FW/BW mismatch.

In general, training quantized DNNs with such a mixture of different weight quantizations and decaying $T$ will not necessarily result in a quantized DNN that uses a low bitwidth. To enforce a low-bit quantized DNN, we therefore add to the loss function a separate regularizer for each layer

$$r(\boldsymbol{\alpha}) = \frac{\lambda \mathbf{g}^T \mathbf{a}(\boldsymbol{\alpha})}{S}, \tag{6}$$

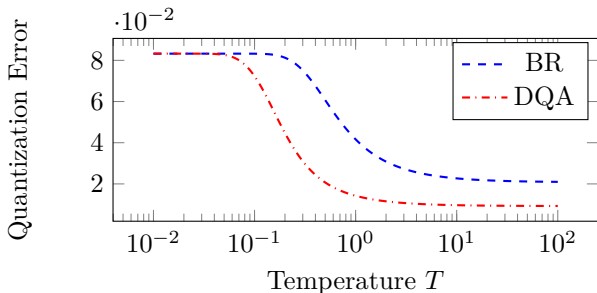

Figure 2: The quantization error of BR and DQA for different temperatures $T$. In particular, a low temperature corresponds to 2bit quantization, while a high temperature means a mixture of float32 and 2bit quantization or a mixture of 2/4/8bit quantization, respectively.

where $S$ is the number of weights in the whole network. Note, that the normalization by $S$ makes the regularizer, and therefore the choice of $\lambda$, independent of the actual network size. $\mathbf{g} = [g_1, g_2, \cdots, g_K]^T$ is a penalty vector, where $g_k$ is increasing with growing $k$. Because we assume, that the quantization functions $Q_k(\mathbf{w}; n_k)$ are sorted by the bitwidth, i.e., $n_1 < n_2 < ... < n_K$ adding $\mathbf{g}^T \mathbf{a}(\boldsymbol{\alpha})$ penalizes if large attention values are assigned to quantizers with a large bitwidth. Hence, it helps the method to converge to the lowest-bit quantization. Algorithm 1 summarizes the DQA training. To quantize a given value $x$, we use min-max, Statistics-Aware Weight Binning SAWB, Binary Weight Network (BWN) or Ternary Weight Network (TWN) as defined by Nikolić et al. (2020); Choi et al. (2018); Rastegari et al. (2016); Li et al. (2016a) respectively, and detailed in appendix A.1

---

**Algorithm 1** DQA algorithm for a single network layer

---

Inputs: Input vector $\mathbf{x}$, initial softmax temperature $T(0)$, final softmax temperature $T(B)$, number of training iterations $B$, and layer transfer function $f$
Output: Output tensor $\mathbf{y}$

$\psi = e^{\frac{log\left(\frac{T(B)}{T(0)}\right)}{B}} < 1$
for each $b = 1, 2, ..., B$ do
    $T(b) \leftarrow T(0)\psi^b$
    $\boldsymbol{\alpha} \leftarrow \frac{\boldsymbol{\alpha}}{\text{std}(\boldsymbol{\alpha})}$
    $\mathbf{a} \leftarrow \text{softmax}(\boldsymbol{\alpha}/T(b))$
    $\mathbf{q} = \mathbf{Q}^T \mathbf{a}$ (linear combination)
    $\mathbf{y} = f(\mathbf{x}, \mathbf{q})$
    Update $\mathbf{w}$ and $\boldsymbol{\alpha}$ via backpropagation.
end for

---

## 4 Experiments

In this section we will first introduce the benchmark protocol that we use to evaluate our method, then we report different results obtained by DQA and compare them with other training procedures.

### 4.1 Benchmark Protocol

To evaluate our method DNN Quantization with Attention (DQA), we perform experiments on the three object recognition datasets CIFAR10, CIFAR100 and ImageNet ILSVRC 2012. For each dataset, we use DQA to train low-bit quantized versions of the ResNet18 (He et al. (2016)) and MobileNetV2 (Sandler et al. (2018)) network architectures. Low-bit means, that we consider networks that only use $n = 1$ or $n = 2$bit for quantization.

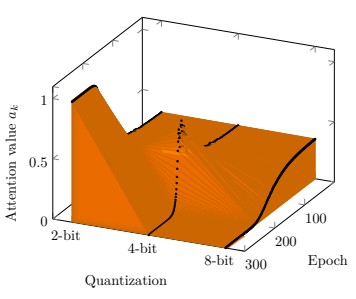
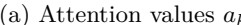

(a) Attention values $a_k$

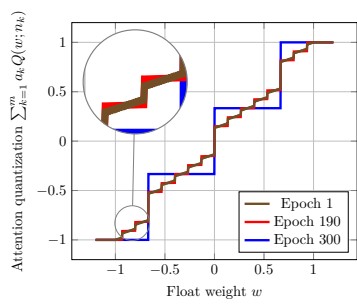

(b) Resulting quantization function

Figure 3: The evolution of the attention values $a_k$ and the resulting quantization function for the first layer of a quantized ResNet18 trained on CIFAR100.

For CIFAR10 and CIFAR100, we start from randomly initialized parameters $\mathbf{w}$ and train the quantized networks for 300 epochs. As an optimizer, we use SGD with an initial learning rate $\gamma = 0.1$, which is divided by 10 every 100 epochs. The training batch size is 128.

On the ImageNet ILSVRC 2012 dataset, we train the quantized networks for 90 epochs, using a batch size of 256 images. As an initial learning rate, we again use $\gamma = 0.1$ which is divided by 10 every 30 epochs. This way, we again apply two equally spaced learning rate drops over the full 90 epochs.

For all our experiments, we either quantize only the weights or both weights and activations using DQA with three different quantization functions $\{Q_1, Q_2, Q_3\}$. More specifically, we either consider a mixture of three min-max quantization functions that use $n_1 = 2$bit, $n_2 = 4$bit and $n_3 = 8$bit, respectively or a mixture of BWN, TWN and 8bit min-max quantization. For all experiments, we use an exponential temperature schedule with an initial temperature $T(0) = 100$ that is cooled down to the final value of $T(B) = 0.03$. The soft attention vector is initialized according to

$$\alpha_k = \frac{\sum_{j=1, j \neq k}^{N} n_j}{\sum_{j=1}^{N} n_j}. \tag{7}$$

Note, that since the quantization functions $Q_k(\mathbf{w}; n_k)$ are assumed to be sorted by the bitwidth, i.e. $n_1 < n_2 < \cdots < n_K$, this initialization assigns the highest attention to the quantization function with the lowest bitwidth. The initialization therefore acts as a prior that favours low-bit quantized DNNs and helps us to converge to small bit widths early during training. To further encourage low-bit quantized DNNs, we use the penalty values $\mathbf{g} = [1, 4, 16]^T$ that penalize quantization functions with a large bitwidth. Note, that we always compare networks that are quantized to the same bitwidth and thus have the same memory footprint and the same computational complexity.

### 4.2 Results

In the first experiments, we report the obtained accuracy achieved by DQA and compare it to three different baselines: 1) The full-precision network with float32 parameters. 2) The quantized network that uses 2bit quantized parameters and uses vanilla training without any relaxation scheme. 3) Binary-Relax (BR). To have a fair comparison to BR, we also report some results where we consider BR with the same mixture of quantization functions, i.e.,

$$\mathbf{q} = \frac{\omega Q_1(\mathbf{w}, n_1) + Q_2(\mathbf{w}, n_2) + Q_3(\mathbf{w}, n_3)}{\omega + 2}, \tag{8}$$

where $\omega$ is initialised to 1 and multiplied by 1.02 after each epoch. In other words, we use a fixed schedule to move from 8bit to 2bit quantization.

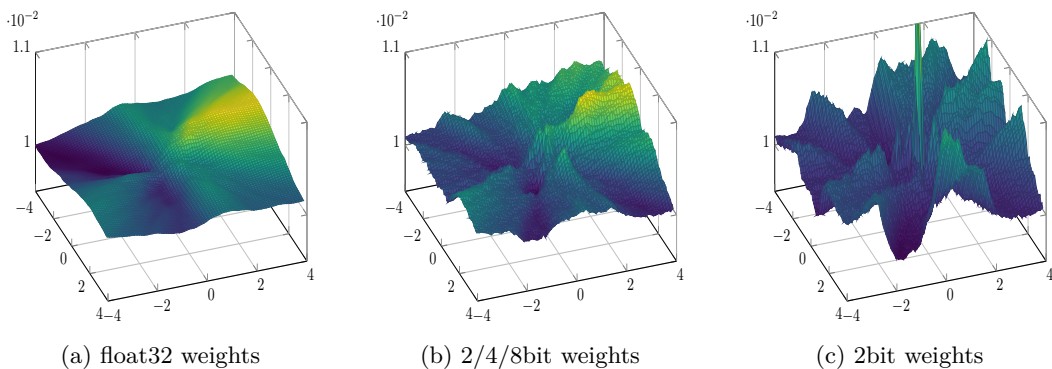

(a) float32 weights        (b) 2/4/8bit weights        (c) 2bit weights

Figure 4: The loss surfaces for a randomly initialized ResNet18, with float32, 2bit or mixed 2,4,8bit quantized weights, evaluated on the CIFAR10 dataset. 2bit quantized weights yield a non-smooth loss surface with many local minima, what is undesirable for optimization. Mixing 2,4,8bit quantizers alleviates this problem.

Table 1 and 2 show the experimental results for the CIFAR10 and CIFAR100 datasets when we quantize only weights and when we quantize both weights and activations, respectively. We report the final validation accuracy of the quantized DNNs for different choices of the quantization functions $\{Q_1, Q_2, Q_3\}$. In general, all reported validation accuracies are the result of a single training run. Only for the experiments that use BWN quantization, we report the average validation accuracy computed over 5 runs, because the convergence of BWN quantized networks proved to be noisy, which shadowed the effects of DQA. Our proposed method archives accuracies that are comparable to the full-precision baseline, while outperforming the 2bit quantized baseline and BR.

The second experiment aims at studying the behavior of the attention values $a_k$ during training. Figure 3 shows the evolution of the attention values $a_k$ and the corresponding quantization function. We can observe from 3a that all attention values are equal at the beginning of the training but – due to the penalty term and the temperature schedule – they slowly converge towards a maximum attention value for the 2bit quantization. This evolution can also be seen in 3b where we show how the resulting quantization function evolves during training. Note, that the quantization function is smoothed out at the beginning and converges more and more towards the 2bit quantization curve at the end of the training. This smooth transition is the reason why DQA yields better results than training with just a single fixed quantization.

Interestingly, compared to a single low-bit quantization, DQA yields smoother loss surfaces. Figure 4 visualizes the loss surface of a ResNet18 with randomly initialized weights on the CIFAR10 dataset. Here, we apply the method proposed by Li et al. (2017) that samples two random directions in the parameter space of a DNN and visualizes the loss along these directions. Obviously, the loss surface is the smoothest for a float32 network. In comparison, the same ResNet18 with 2bit min-max quantized weights yields a very rough loss surface. For the 2bit case, optimization can get stuck easily in one of the numerous local minima. Moreover, it yields gradients that change quickly in direction and magnitude, causing severe oscillations and effectively making the training unstable. However, if we apply DQA and use a mixture of 2, 4 and 8bit min-max quantization to quantize the network weights, the loss surface is smoothed out. Therefore, compared to DNNs that are trained with only one low-bit quantization scheme, quantized DNNs trained with DQA typically converge faster at the beginning of the training and reach a better final optimum.

The third experiment compares DQA with other methods for quantized DNNs trained on the ImageNet ILSVRC 2012 dataset. Table 3 (parameter quantization only) and Table 4 (parameter and activation quantization) show that DQA outperforms the quantized 2bit baseline and BR when considering different quantization approaches. Moreover, DQA causes a significantly smaller drop in accuracy when quantizing MobileNetV2. Thus, it may represent a

| | Data | $n_1$ | $Q_1$ | $n_2$ | $Q_2$ | $n_3$ | $Q_3$ | $\lambda$ | Acc |
|---|---|---|---|---|---|---|---|---|---|
| R18 | C10 | 32 | FP | - | - | - | - | - | 95.2% |
| R18 | C10 | 2 | min-max | - | - | - | - | - | 91.5% |
| R18+BR | C10 | 2 | min-max | 32 | FP | - | - | - | 93.0% |
| R18+BR | C10 | 2 | min-max | 4 | min-max | 8 | min-max | - | 93.7% |
| R18+Ours | C10 | 2 | min-max | 4 | min-max | 8 | min-max | 5 | **94.8**% |
| R18 | C10 | 2 | SAWB | - | - | - | - | - | 94.8% |
| R18+BR | C10 | 2 | SAWB | 4 | SAWB | 8 | SAWB | - | 95.1% |
| R18+Ours | C10 | 2 | SAWB | 4 | SAWB | 8 | SAWB | 1 | **95.4**% |
| R18 | C10 | 1 | BWN | - | - | - | - | - | 93.8% |
| R18+BR | C10 | 1 | BWN | 2 | TWN | 32 | FP | - | 94.2% |
| R18+Ours | C10 | 1 | BWN | 2 | TWN | 32 | FP | 5 | **94.5**% |
| R18 | C10 | 2 | TWN | - | - | - | - | - | 94.3% |
| R18+BR | C10 | 2 | TWN | 4 | min-max | 8 | min-max | - | 94.5% |
| R18+Ours | C10 | 2 | TWN | 4 | min-max | 8 | min-max | - | **94.8**% |
| R18 | C100 | 32 | FP | - | - | - | - | - | 77.9% |
| R18 | C100 | 2 | min-max | - | - | - | - | - | 70.0% |
| R18+BR | C100 | 2 | min-max | 32 | FP | - | - | - | 72.9% |
| R18+BR | C100 | 2 | min-max | 4 | min-max | 8 | min-max | - | 74.0% |
| R18+Ours | C100 | 2 | min-max | 4 | min-max | 8 | min-max | 10 | **76.4**% |
| R18 | C100 | 2 | SAWB | - | - | - | - | - | 77.0% |
| R18+BR | C100 | 2 | SAWB | 4 | SAWB | 8 | SAWB | - | 77.3% |
| R18+Ours | C100 | 2 | SAWB | 4 | SAWB | 8 | SAWB | 5 | **78.1**% |
| R18 | C100 | 1 | BWN | - | - | - | - | - | 75.0% |
| R18+BR | C100 | 1 | BWN | 2 | TWN | 32 | FP | - | 75.3% |
| R18+Ours | C100 | 1 | BWN | 2 | TWN | 32 | FP | 30 | **75.9**% |
| R18 | C100 | 2 | TWN | - | - | - | - | - | 76.1% |
| R18+BR | C100 | 2 | TWN | 4 | min-max | 8 | min-max | - | 76.3% |
| R18+Ours | C100 | 2 | TWN | 4 | min-max | 8 | min-max | 20 | **76.7**% |

Table 1: Obtained accuracy of ResNet18 (R18) trained on CIFAR10 (C10) and CIFAR100 (C100) for quantized weights, only. We consider numerous quantization functions (min-max, SAWB, BWN and TWN). Note, that FP refers to full precision (i.e. $Q(\mathbf{w}, 32) = \mathbf{w}$).

promising training procedure that makes existing quantization methods more powerful and, hence, helps us to train lightweight DNN architectures.

## 5 Conclusion

In this paper, we introduced DQA, a novel learning procedure for training low-bit quantized DNNs, using existing quantization methods. Instead of using only a single quantization precision during training, DQA relaxes the problem and uses a mixture of high, medium and low-bit quantization functions. Our experiments on popular object recognition datasets, such as CIFAR10, CIFAR100 and ImageNet ILSVRC 2012, show that DQA can be used to train highly accurate low-bit quantized DNNs that achieve a good accuracy compared with state-of-the-art counterparts.

If we compare to the full-precision networks, DQA yields a significantly lower accuracy drop than other training procedures that only use a single quantization precision and bitwidth during training, This is especially true when quantizing DNN architectures that are already designed to be lightweight and efficient, such as the MobileNetV2. Because such architectures are already small, they are naturally harder to compress.

DQA also compares favourably to Binary-Relax (BR), another training procedure for quantized DNNs that applies a mixture of quantized and full-precision weights during training.

|       | Data | $n_1$ | $Q_1$   | $n_2$ | $Q_2$   | $n_3$ | $Q_3$   | $\lambda$ | Acc       |
|-------|------|-------|---------|-------|---------|-------|---------|-----------|-----------|
| R18   | C10  | 32    | FP      | -     | -       | -     | -       | -         | 95.2%     |
| R18      | C10  | 2  | min-max | -  | -       | -  | -       | -  | 87.8%     |
| R18+BR   | C10  | 2  | min-max | 32 | FP      | -  | -       | -  | 89.5%     |
| R18+Ours | C10  | 2  | min-max | 4  | min-max | 8  | min-max | 5  | **90.4**% |
| R18      | C10  | 2  | PS      | -  | -       | -  | -       | -  | 94.4%     |
| R18+BR   | C10  | 2  | PS      | 32 | FP      | -  | -       | -  | 94.3%     |
| R18+Ours | C10  | 2  | PS      | 4  | PS      | 8  | PS      | 1  | **94.7**% |
| R18      | C100 | 32 | FP      | -  | -       | -  | -       | -  | 77.9%     |
| R18+BR   | C100 | 2  | min-max | 32 | FP      | -  | -       | -  | 65.2%     |
| R18+Ours | C100 | 2  | min-max | 4  | min-max | 8  | min-max | 10 | **68.3**% |
| R18      | C100 | 2  | PS      | -  | -       | -  | -       | -  | 75.2%     |
| R18+BR   | C100 | 2  | PS      | 32 | FP      | -  | -       | -  | 75.9%     |
| R18+Ours | C100 | 2  | PS      | 4  | PS      | 8  | PS      | 5  | **78.1**% |

Table 2: Obtained accuracy of ResNet18 (R18) trained on CIFAR10 (C10) and CIFAR100 (C100), when quantizing both weights and activations to 2bit. Note, that PS refers to PACT-SAWB.

|          | $n_1$ | $Q_1$   | $n_2$ | $Q_2$   | $n_3$ | $Q_3$   | $\lambda$ | Top-1 (Top-5)       |
|----------|-------|---------|-------|---------|-------|---------|-----------|---------------------|
| R18      | 32 | FP      | -  | -       | -  | -       | -  | 69.9% (89.1%)         |
| R18      | 2  | min-max | -  | -       | -  | -       | -  | 58.7% (81.9%)         |
| R18+Ours | 2  | min-max | 4  | min-max | 8  | min-max | 1  | **66.9**% (**87.4**%) |
| MV2      | 32 | FP      | -  | -       | -  | -       | -  | 69.0% (89.0%)         |
| MV2      | 2  | min-max | -  | -       | -  | -       | -  | 44.2% (69.8%)         |
| MV2+Ours | 2  | min-max | 4  | min-max | 8  | min-max | 1  | **52.2**% (**77.1**%) |
| R18      | 1  | BWN     | -  | -       | -  | -       | -  | 61.0% (83.5%)         |
| R18+Ours | 1  | BWN     | 2  | TWN     | 8  | min-max | 10 | **61.4**% (**83.7**%) |

Table 3: Experiments on the ImageNet dataset, using the ResNet18 (R18) and the Mo-bileNetV2 (MV2) networks with quantized weights, only. Quantized DNNs trained with DQA consistently outperform quantized DNNs that have been trained with just a single quantization method. It also drastically reduces the accuracy drop when quantizing MobilenetV2.

|          | $n_1$ | $Q_1$   | $n_2$ | $Q_2$ | $n_3$ | $Q_3$   | $\lambda$ | Top-1 (Top-5)       |
|----------|-------|---------|-------|-------|-------|---------|-----------|---------------------|
| R18      | 32 | FP      | -  | -     | -  | -       | -   | 69.9% (89.1%)         |
| R18      | 2  | min-max | -  | -     | -  | -       | -   | 40.7% (69.9%)         |
| R18+BR   | 2  | min-max | 32 | FP    | -  | -       | -   | 57.7% (81.5%)         |
| R18+Ours | 2  | min-max | 4  | min-max | 8 | min-max | 0.5 | **60.4**% (**83.4**%) |

Table 4: Experiments on the ImageNet dataset, when quantizing both weights and activations of ResNet18 (R18).

.

However, while BR uses a fixed scheme to mix the network weights of different precision, DQA can learn how to mix them in an optimal way and how to gradually move from high precision to low precision. In practice, this helps training and results in quantized DNNs with higher accuracy.

DQA is agnostic to and can be used with many different existing quantization methods, such as min-max, PACT-SAWB, Binary-Weight and Ternary-Weight quantization. Therefore, DQA is a very promising extension to existing DNN quantization methods.

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

## A Appendix

### A.1 Choosing the Quantization Functions

Quantization describes the process of representing a value $x \in \mathcal{X}$ with a corresponding quantized value $q \in \mathcal{Q}$, using a quantization function $Q : \mathcal{X} \to \mathcal{Q}$. Here, $\mathcal{Q} = \{q_1, q_2, ..., q_{2^n}\}$ is the set of quantization steps that is much smaller than $\mathcal{X}$, i.e., $|\mathcal{Q}| << |\mathcal{X}|$. For a given $w$ and $\mathcal{Q}$, the quantization function minimizes the distance between $w$ and $q$, i.e.,

$$Q(x; n) = \arg\min_{q \in \mathcal{Q}} \|x - q\|, \tag{9}$$

where $\| \cdot \|$ is the Euclidean norm. There are different methods how to construct $\mathcal{Q}$ that yield different quantization schemes, like uniform or non-uniform quantization.

The first method we may consider is the one introduced in Nikolić et al. (2020). For $\mathcal{X} = [x_{min}, x_{max}]$ they define

$$q_i = x_{min} + (i - 1)\frac{x_{max} - x_{min}}{2^n - 1}, \ \ i = 1, 2, ..., 2^n. \tag{10}$$

In particular, the values $q_i$ are uniformly distributed between the values $x_{min}$ and $x_{max}$, what is known as min-max quantization.

The second method we use with our proposed training procedure is Statistics-Aware Weight Binning (SAWB) Choi et al. (2018). The quantization values are again distributed uniformly over a given interval. However, instead of using the limits $x_{min}$ and $x_{max}$, SAWB introduces a limit $\alpha$, i.e.,

$$q_i = -\alpha + (i - 1)\frac{2\alpha}{2^n - 1}, \ \ i = 1, 2, ..., 2^n. \tag{11}$$

The optimal $\alpha$ can be calculated in a calibration step, using data. In particular, we minimize the mean-square quantization error

$$\alpha^* = \arg\min_{\alpha} \mathrm{E}_{x \sim p(x)}[\|x - Q(x; n, \alpha)\|^2] \tag{12}$$

with respect to $\alpha$. After calibration, we can use $Q(x; n) = Q(x; n, \alpha = \alpha^*)$ for quantization.

For both min-max and SAWB quantization, the solution of Eq. equation 9 is straight-forward to obtain. It is a uniform quantization function with equally spaced quantization steps that is defined by

$$Q(w; n) = \begin{cases} q_1 & , x <= q_1 \\ q_1 + \frac{q_{2^n} - q_1}{2^n - 1}\mathrm{round}\left(x\frac{2^n - 1}{q_{2^n} - q_1}\right) & , \mathrm{others} \\ q_{2^n} & , x > q_{2^n} \end{cases}. \tag{13}$$

Another quantization function worth to mention was introduced for the Binary Weight Network (BWN) Rastegari et al. (2016). It uses a scaling factor $\beta = \mathbf{E}(|x|)$ and constrains

the quantized values to be binary ($n = 1$). In particular, with $\mathcal{Q} = \{-\beta, \beta\}$, the quantization function is defined as

$$Q(w, 1) = \beta_w \cdot sign(x) = \begin{cases} \beta & , x \geqslant 0 \\ -\beta & , \text{others} \end{cases} \tag{14}$$

In the same vein, Ternary Weight Network (TWN) Li et al. (2016a) introduces a third quantization step to improve the accuracy. A TWN uses a bitwidth of $n = 2$ and a symmetric $\mathcal{Q} = \{-\beta, 0, \beta\}$. Similar to the BWN, the range parameter $\beta$ is calibrated with data. More specifically, we can compute the optimal range $\beta^* = \mathrm{E}_{x\ p(x||x|>\delta)}[|x|]$, where $\delta = 0.7\mathrm{E}[|x|]$ is the symmetric threshold that is used for quantization during calibration. The resulting quantization function is defined as

$$Q(x; 2) = \begin{cases} -\beta & , x \leqslant -\delta \\ 0 & , |x| \leqslant \delta \\ \beta & , x > \delta \end{cases} . \tag{15}$$

