# OpenReview forum: "DNN Quantization with Attention"
_ICLR.cc/2022/Conference — ICLR 2022 Submitted_

### Official Review · Reviewer_1Jav · 2021-10-31

**Correctness:** 3
**Technical Novelty And Significance:** 2
**Empirical Novelty And Significance:** 2
**Recommendation:** 3
**Confidence:** 4

**Main Review:**

Strength:
-	Authors shows the proposed multi-biwidth training effectively reduces quantization error and helps smooth loss landscape with some sample cases and visualizations.

Weaknesses:
-	Overall performance of the given approach is not satisfying. Most recent quantization papers mainly conduct experiments on the large-scale ImageNet dataset since the CIFAR datasets are prone to easy overfitting. Almost all papers I know doing low-bit quantization has better results than this one.

On weight-only 2-bit quantization with MobilenetV2:
SAT: 66.8 (Neural Network Quantization with Scale-Adjusted Training BMVC 2020)
DeepComp 58.1 (Deep compression: Compressing deep neural networks with pruning, trained quantization and huffman coding. ICLR 2016)
This work: 52.2

On both weight and activation 2-bit quantization with ResNet18:
PACT 64.4 (Pact: Parameterized clipping activation for quantized neural networks. arXiv 18)
LQNet 64.9 (Lq-nets: Learned quantization for highly accurate and compact deep neural networks. ECCV 18)
SAT 65.5 (Neural Network Quantization with Scale-Adjusted Training BMVC 2020)
This work: 60.4

-	Some technical details are not clear. The authors use a penalty term (equation 6) to regularize the attention weights of different bitwdith. However, it is not known whether all bitwidths in the network will be converged to the lowest bit which is the target. If some bitwidths are not property converged, will there be any issue on the performance?
-	Writing needs improvement. There are a lot of grammar errors and typos:

Page 2 last paragraph: a way how to train a (delete how)

Page 3 background 4th paragraph: Note, that because (no comma)

Page 3 Background 4th paragraph: this problem is most pronounced for low bitwidhts (typo)


**Summary Of The Paper:**

This work presents a training method for low-bit network quantization. While training, it employs a multi-bitwidth paradigm in order to alleviate the nonsmooth optimization landscape with lower bitwidth. It uses a temperature parameter and a penalty term to force the network to gradually converge to the target low bit. Experiments are conducted on CIFAR10, CIFAR100, ImageNet Classification with ResNet 18 and MobilenetV2.

**Summary Of The Review:**

Overall, this work presents a new approach to help improve the training convergence of low-bit quantization for neural network. Due to the weak results on large-scale datasets and unclear technical details, I do not think it meets our bar at ICLR.

---

> ### Author Response · Authors · 2021-11-19
> **Official comments**
>
> We thank the reviewer for his effort in reading and understanding the proposed method, and his comments that are relevant to us, and help us to improve the paper quality.
>
> The proposed method is a way to use already existing quantization methods and improve them, and not a quantization technique by itself. Therefore, the comparison is done with a given quantization method, and the same quantization method when considering our proposed method. We also compare with Binary Relax which is another method that considers existing quantization techniques and improves them.
>
> More details about the proposed method and its purpose, and corrections are introduced in the final version.

---

### Official Review · Reviewer_H2zp · 2021-11-02

**Correctness:** 2
**Technical Novelty And Significance:** 2
**Empirical Novelty And Significance:** 2
**Recommendation:** 3
**Confidence:** 2

**Main Review:**

Pros:
- The paper is well written and the idea is easy to follow.
- Extensive ablation studies are provided to evaluate different components of the proposed method.

Cons:
- More parameters are introduced in the training stage, such as α. This will increase the computation and storage cost. More theoretical and experimental analysis should be given to study that.
- Multiple quantizers with different bit-width are conducted in the proposed method, which will increase the storage and computation cost for quantization.
- In the experiments, the authors compare their method with the corresponding counterparts with the same bit of n1. However, the proposed method has three quantizers with different bit-width, and n1 is the lowest bit-width. Therefore, this comparison seems unfair. For a fair comparison, the baselines and the proposed method should be compared under the same computation and storage cost.
- The proposed method has not been compared with state-of-the-art approaches, which cannot comprehensively evaluate the proposed method.

**Summary Of The Paper:**

This paper attempt to address a challenging quantization problem, i.e., low-bit quantization. This work utilizes a learnable linear combination of high, medium, and low-bit quantization at the beginning while converging to a single low-bit quantization at the end of the training. In the quantization procedure, multiple quantizers and the corresponding attention matrices are adopted to fuse the quantized weights or activations.

**Summary Of The Review:**

This work utilizes a learnable linear combination of high, medium, and low-bit quantization at the beginning while converging to a single low-bit quantization at the end of the training. In the quantization procedure, multiple quantizers and the corresponding attention matrices are adopted to fuse the quantized weights or activations, which will increase the computation and storage cost. Some experiments are conducted to evaluate the proposed method. However, it lacks some comprehensive and fair comparison.

---

> ### Author Response · Authors · 2021-11-19
> **Official comments**
>
> First of all, we would like to thank the reviewer for pointing out very interesting comments that will help us to improve the paper quality.
>
> Indeed more parameters and quantizers are introduced during training that increases the memory footprint and computational power. However, in this paper, we aim at quantizing a given network for the inference process, and with the proposed method we end up with a quantized network without any additional parameter. Thus the method aims at improving a quantized network performance without adding additional complexity during inference. That will be better explained in the final version.
>
> The comparison is done when considering the inference process when the obtained network using our method is quantized on only n1 bits and thus for inference and deployment it uses the same memory and computational power as other counterparts. More details will be introduced in the Results section to avoid such confusion.
>
> The proposed method is not a quantization technique but a way to better use and improve existing quantization methods. It has been compared to Binary Relax because as the proposed method, it aims to relax the quantization problem and improve already existing quantization techniques. Other state-of-the-art will be added in the final version of the paper to show that when applying our proposed contribution, such methods can be improved.

---

### Official Review · Reviewer_UAi4 · 2021-11-03

**Correctness:** 3
**Technical Novelty And Significance:** 1
**Empirical Novelty And Significance:** 1
**Recommendation:** 3
**Confidence:** 4

**Main Review:**


1) In my understanding, what the paper call attention is simply a value weighting the importance of the different quantization functions (see Eqn. 2-3). Therefore this terminology is misleading in my opinion, as the vector of attention a only depends on the “trainable parameter alpha”, and not does not depend on the input (either or activation) as one would imagine with this name. Maybe I misunderstood something, but this is what I infer from Eqn (4). Eqn (2-3) and Figure 1.

2) The paper does not consier in the literature review techniques for quantizing neural networks [A,B,C] that, to my knowledge, are state-of-the-art for quantizing popular neural networks.
[A] Permute, Quantize, and Fine-tune: Efficient Compression of Neural Networks, Martinez et al., CVPR’2021
[B] And the bit goes down: Revisiting the quantization of neural networks, Stock et al. ICLR’2020
[C] Training with Quantization Noise for Extreme Model Compression, Fan et al. ICLR’2021
At a high level, some elements are similar to these approaches, even though the details may differ.  One noticeably similarity is the various of precision, which is already present in the work by Fan et al [C] when they consider both blocks that are not quantized (i.e., 32 bits) versus some that are quantized with low-precision, and randomly choose. In my own experience, choosing randomly a choice is better that relaxation, so I guess the paper should have included such a comparison. In any case the paper should be better positioned against the recent literature.

3) The paper is compared to poor baselines. As a result, the paper reports results that do not look competitive the state of the art [A,B,C] on Imagenet ILSVRC 2012, which is the most significant benchmark in the paper. For instance, Stock et al. [B] report some results with R18  (Figure 3 left, table 3) that are as follow: compression factors of x20 for top-1 accuracy at 67.87. In the submitted paper. Additionally, more recent papers [A] and [C] are compared to [B] on larger networks and show that they further improve results. Therefore I conclude that the proposed approach is not competitive, while additionally requiring a more engaged scheduling that may not generalize as well to other training settings. The paper states that it could be combined with any compression method, therefore in this context it would be worth using the same or similar quantization as in [A,B,C] and show how the method compares to these methods.

4) Formally I have nothing against having in the same section the introduction and the related work. In the case of this paper, I found that the introduction is actually more a related work than a formal introduction providing the motivation and rationale of the paper content at the core of the initial discussion.  This discussion appears later in the background section. While I know well this area and therefore the problem at stake, I would advise to re-work jointly these two sections.

5) The paper  mentions that the method could be used for quantization activation, but only addresses the case of weight quantization. I think that a lot of practical considerations would appear with activation quantization, so I would suggest either to support this claim with experiments,  or to suppress or soften this unsupported claim.

6) The paper needs some polishing. Some mistakes indicate that the paper was not analyzed by a spell-checker, for instance in the abstract: conterparts -> counterparts. The term bitwidth is not established (and occurred at least once with a typo).




**Summary Of The Paper:**

The papers addresses the problem of compression of neural networks. The paper builds upon binary-relax prior work. The precision is adapted during training with a mixture-based quantization method through temperature cooling and a set of an ``attention’’ vector a. The idea of the method, called DQA, is to progressively moves from a mixture of quantization functions (mixing high with low precision, for instance 32 bit with 2 bit training), to a single one (low precision) towards the end of the training. The paper states that the method can be used with several types of quantization methods. The evaluation is carried on computer vision architectures (ResNet18, MobileNet) for image recognition tasks (Cifar-10, Cifar-100, Imagenet ILSVRC 2012).


**Summary Of The Review:**

The paper does not demonstrate that the method is a significant contribution to the state of the art in quantizing neural networks. The experiments are only applied on image classification task with small architectures, and in the setting that I found comparable in the literature, the results do not look great, which questions the significance of this work.

---

> ### Author Response · Authors · 2021-11-19
> **Official Answer**
>
> We would like to thank the reviewer for his comments that help us to improve the paper quality.
>
> Some typos in the paper make some parts a bit confusing. The final version will be better presented.
>
> The paper aims at finding a way to improve quantization methods and not at introducing a new one. We used some well known quantization methods to show that. More quantization methods will be added in the final version to show how they can be improved using our method.
>
> We are separating the introduction and related work to make each part more relevant and clear in the final version.
>
> In the section Results, Tables 2 and 4 report obtained accuracy when both weights and activations are quantized.
>
> Also in the final version, we are considering more complex tasks as object detection to more stress the proposed contribution.

---

### Official Review · Reviewer_4bsX · 2021-11-08

**Correctness:** 3
**Technical Novelty And Significance:** 2
**Empirical Novelty And Significance:** 2
**Recommendation:** 3
**Confidence:** 4

**Main Review:**

Strengths:
1. According to the experiment, DOA performs better than the naive quantization strategy and Binary-Relax consistently across the experimented datasets and networks.

Weaknesses:
1. The presentation needs to be improved. There are grammar errors and typos in the paper.
2. The paper compares the performance of DOA with only one related quantization work (Binary-Relax). It is not sufficient to demonstrate the effectiveness of the proposed work. There are many quantization works, both quantization-aware training and post-training quantization. It seems that some of them may have better performance in the experimented settings. For example, the work LQ-Nets reports an accuracy of 68% with 2-bit weights and 32 bits activation with ResNet-18 on ImageNet, but DOA proposed in this paper only achieves an accuracy of 66.9%.

Minor comments or questions:
1. How to decide what quantization method to use (e.g., min-max, SAWB, BWN, TWN) when using DAQ in practice? Appendix A only defines each quantization method but doesn't give any guidance on how to choose them.
2. For experiments on the ImageNet dataset using ResNet18, why not report R18+BR and MV2+BR results?
3. In section 4.2, the paper mentions all reported validation accuracies are the results of a single training run. It might be better to report averaged results across several runs even if the convergence of the networks is not noisy.
4. In Table 1, it seems that DQA using SWAB consistently gives better results than the FP version. Do you have any insight regarding this?

**Summary Of The Paper:**

This paper proposes DNN quantization with Attention (DQA), which uses a learnable linear combination of high, medium, and low-bit quantization at the beginning. It gradually converges to a single low-bit quantization at the end of training. Experiments show that DQA outperforms the naive quantization and the Binary-Relax method consistently across three datasets and two networks.

**Summary Of The Review:**

Overall, I think the paper is not good enough due to aforementioned nontrivial weaknesses.

---

> ### Author Response · Authors · 2021-11-19
> **Official Answer**
>
> We would like to thank the reviewer for his constructive reviews. The presentation will be improved in the final version and we are checking all grammar errors.
> In this paper, we are comparing to Binary-relax because this method tries to relax the quantization problem and improves already existing quantization methods as our method. The idea here is not to introduce a new quantization method but to find a way how to use the already existing methods more efficiently. For Instance, LQ-Net is a quantization method that can be used in our method to be improved. We will reformulate the method purpose in the paper to make it more clear. We will also add more quantization methods in the final version and show how they can be improved using our method.
>
> Due to time, we could not run more experiments. More results will be added to the final version.

---

### Decision · Program_Chairs · 2022-01-20

**Decision:**

Reject

**Comment:**

This paper proposes a new learning procedure for quantizing neural networks. Basically, DQA method proposed in this paper uses attention to obtain a linear combination of the existing network quantization techniques and uses it to pursue more efficient quantization.

Overall, it seems the submission was written in haste, so there are many typos and errors. Above all, the motivation that it can be applied to various existing techniques could not be proved experimentally at all since it only covers one somehow obsolete work. In addition, as in [1], it seems necessary to quantize not only weights but also activations, or to verify in lightweight networks such as MobileNetV2 rather than ResNet.

[1] Cluster-Promoting Quantization with Bit-Drop for Minimizing Network Quantization Loss, ICCV 2021